The Microphenotron: a novel method for screening plant growth-promoting rhizobacteria

http://orcid.org/0000-0003-2465-2399 Raheem Asif 1 2
http://orcid.org/0000-0002-7686-2702 Ali Basharat 1 basharat.ali.mmg@pu.edu.pk
1 Institute of Microbiology and Molecular Genetics, University of the Punjab , Lahore, Punjab , Pakistan
2 Department of Microbiology, Balochistan University of Information Technology, Engineering and Management Sciences (BUITEMS) Quetta , Quetta, Balochistan , Pakistan
Nelson Craig
Electronic publication date: 2022 May 13
Publication date: 2022
Volume: 10
Electronic Location ID: e13438
Received 2020 Sep 28; Accepted 2022 Apr 22
Copyright: © 2022 Raheem and Ali
Copyright year: 2022
Copyright holder: Raheem and Ali
License: This is an open access article distributed under the terms of the Creative Commons Attribution License, which permits unrestricted use, distribution, reproduction and adaptation in any medium and for any purpose provided that it is properly attributed. For attribution, the original author(s), title, publication source (PeerJ) and either DOI or URL of the article must be cited.
License URL: https://creativecommons.org/licenses/by/4.0/

Keywords: Microphenotyping system, IAA, Arabidopsis thaliana, Rhizobacteria, ACC-deaminase, Mutant lines

Funding: The Higher Education Commission of Pakistan IRSIP No. 1-8/HEC/HRD/2012/2586 and No. 17-5-6 (Bm6-133)/HEC/Sch/2010 The Higher Education Commission of Pakistan providing funding to Asif Raheem under IRSIP No. 1-8/HEC/HRD/2012/2586 and No. 17-5-6 (Bm6-133)/HEC/Sch/2010. The funders had no role in study design, data collection and analysis, decision to publish, or preparation of the manuscript.

==============================
Background

The ‘Microphenotron’ is an automated screening platform that uses 96-well microtiter plates to test the response of seedlings to natural products. This system allows monitoring the phenotypic effect of a large number of small molecules. Here, this model system was used to study the effect of phytohormones produced by plant growth-promoting rhizobacteria (PGPR) on the growth of wild-type and mutant lines of Arabidopsis thaliana.

Methods

In the present study, high-throughput screening based on ‘Microphenotron’ was used to screen PGPRs. Rhizobacteria were isolated from the rhizosphere of Acacia Arabica, which was growing in saline habitats. The phylogeny of these rhizobacteria was determined by 16S rRNA gene sequencing. Strains were screened for plant growth-promoting traits such as auxin production, 1-aminocyclopropane-1-carboxylate (ACC) deaminase activity, and phosphate solubilization. Ultra-Performance Liquid Chromatography (UPLC) was used to detect the presence of different indolic compounds. Finally, PGPR were evaluated to enhance the growth of A. thaliana in the ‘Microphenotron’ system and pot trials.

Results

Selected rhizobacteria strains showed positive results for multiple plant-growth promoting traits. For instance, strain (S-6) of Bacillus endophyticus exhibited the highest ACC-deaminase activity. UPLC analysis indicated the presence of different indolic compounds in bacterial extracts that included indole lactic acid (ILA), indole carboxylic acid (ICA), and indole-3-acetic acid (IAA). Two strains (S-7 and S-11) of Psychrobacter alimentarius produced the most IAA, ICA and ILA. A screening bioassay through 96-well microtiter plates with wild-type Col. N6000 showed an increase in root growth and proliferation. The highest twofold increase was recorded in root growth with B. thuringiensis S-26 and B. thuringiensis S-50. In pot trials, mutant lines of A. thaliana impaired for auxin signaling showed that B. endophyticus S-6, Psy. alimenterius S-11, Enterobacter asburiae S-24 and B. thuringiensis S-26 used auxin signaling for plant growth promotion. Similarly, for ethylene insensitive mutant lines (ein2.5 and etr1), Prolinoborus fasciculus S-3, B. endophyticus S-6, Psy. alimenterius S-7, E. asburiae S-24, and B. thuringiensis S-26 showed the involvement of ethylene signaling. However, the growth promotion pattern for most of the strains indicated the involvement of other mechanisms in enhancing plant growth. The result of Microphenotron assays generally agreed with pot trials with mutant and wild type A. thaliana varieties. Bacterial strains that induced the highest growth response by these cultivars in the ‘Microphenotron’ promoted plant growth in pot trials. This suggests that Microphenotron can accelerate the evaluation of PGPR for agricultural applications.

Introduction

Plant growth-promoting rhizobacteria (PGPR) play a very critical role in soil fertility and can increase plant growth and agricultural productivity. The numbers of bacterial strains of genus Pseudomonas, Bacillus, Enterobacter, Prolinoborus, Psychrobacter, and Brevibacterium promote plant growth. The application of these rhizobacteria significantly increases the growth and yield of agriculturally important crops (Fahad et al., 2015; Nie et al., 2015; Zahid, 2015; Araújo et al., 2020). Therefore, the application of PGPR may help to minimize the use of chemical fertilizers that can also reduce the production cost and environmental risks (Lucy, Reed & Glick, 2004; Khalid, Arshad & Zahir, 2004).

PGPR can be characterized based on beneficial biochemical attributes, such as phytohormones production and enzymes like 1-amino cyclopropane-1-carboxylate (ACC) deaminase (Haque et al., 2020). Exogenous application of IAA of microbial origin can also trigger growth response in inoculated plants (Ali, Sabri & Hasnain, 2010). Microorganisms can produce auxin in pure culture and in the rhizosphere (Barea et al., 2005; Rajamanickam et al., 2018). Auxin production in the rhizosphere could provoke a physiological response in the host plant. Screening of rhizobacteria with one or few PGPR traits is a basic criterion for the selection of bacteria. For agricultural applications, screening of rhizobacteria based on auxin production has provided a reliable tool for the selection of PGPR (Ali et al., 2009). We have also reported the plant growth enhancement by auxin producing PGPR (Aslam & Ali, 2018; Raheem et al., 2018).

A variety of techniques have been used to screen rhizobacteria for traits associated with PGPR that do not consistently predict a desirable outcome for plant growth and productivity. Similarly, the application of potential PGPR that performs well in a greenhouse often fails to deliver expected results in field settings (Basu et al., 2021). Screening and selection of suitable PGPR candidates in pot trials is a time taking process and an incomplete understanding of the mechanisms of plant growth promotion hinders the PGPR applications in the field. Natural variations in bacterial traits and abiotic factors can also influence microbial performance in the soil (Joshi et al., 2019). Therefore, we need reliable high throughput and low-cost bioassays to screen a large number of bacteria to identify effective PGPR.

The Microphenotron is an automated screening platform that has been used to evaluate the effect of different chemical treatments on A. thaliana (Burrell et al., 2017). This system uses 96-well microtitre plates to deliver chemical treatments to seedlings and can readily demonstrate the effect of metabolites by triggering major changes in root architecture in A. thaliana (Forde et al., 2013). Moreover, mutant lines of A. thaliana can be used to study different signaling pathways (Caviglia et al., 2018). For example, the GUS reporter gene has been used to study the α-glucuronidase induced by microbial indole-3-acetic acid (IAA) in histochemical analysis. Knockouts of the functional version of a gene provide another way to look deep at the molecular level which facilitates by A. thaliana or tomato mutant lines (Jiang et al., 2013; Martínez et al., 2020). Microorganisms also produce a variety of metabolites that can influence plant growth and productivity. This system may also be used to screen PGPR as it allows direct interaction of selected microorganisms with the root system of plants, but the Microphenotron has not been evaluated as a method to select effective PGPR.

To the best of our knowledge, we are first time reporting the use of ‘Microphenotron’ to screen PGPR for plant growth promotion. In the current study, rhizobacteria were isolated from the rhizosphere of halotolerant plants growing in the Pothohar salt range, Pakistan. Strains were tested with in vitro assays for traits associated with plant growth promotion. Finally, the strains were tested in both the Microphenotron and pot trials with wild type and knockout mutants of A. thaliana.

Materials and Methods

Isolation of rhizobacteria

Bacterial strains were isolated from the rhizospheric soil of Acacia Arabica (L.) that inhabited the saline areas of the Pothohar salt range, Pakistan. The rhizospheric soil samples were collected in clean bags. Samples were processed within 24 h by making serial dilutions in sterilized distilled water and inoculated on L-agar (Luria-Bertani) plates supplemented with four different NaCl concentrations i.e., 0.25, 0.5, 0.75 and 1 M. Pure cultures were obtained by quadrant streaking after picking distinct colonies from inoculated plates.

16S rRNA gene sequencing

The taxonomic status of bacterial strains was confirmed by 16S rRNA gene sequencing. DNA was isolated from freshly grown bacterial cultures by using the Genomic DNA Purification Kit (Promega, Madison, WI, USA) according to the manufacturer’s instructions. A fragment (1.5-kb) of the 16S rRNA gene was amplified by using forward and reverse primers (Johnson, 1994). PCR amplification was accomplished in Thermocycler Primus 96 (PeQLab, Erlangen, BY, Germany) as described previously (Raheem et al., 2018). The amplified PCR products were purified by using the QIAquick Gel Extraction Kit (Venlo, Netherlands). Samples were sequenced by using 27f and 1522r primers by sending purified PCR products to Eurofins, United Kingdom (UK). For final identification, sequences of the 16S rRNA gene were compared with already deposited sequences in the GenBank through BLAST analysis. Phylogenetic relationships among bacterial strains were determined by aligning sequences using the Neighbor-Joining method with 1,000 bootstrap replicates. The tree was plotted using MEGA 7 software.

Auxin production and bacterial growth

Auxin production ability of bacterial isolates was determined in the presence and absence of precursor L-tryptophan. Strains were grown in 20 ml L-broth medium supplemented with 0.5, 1, 1.5 and 2 M NaCl in 50 ml Erlenmeyer flasks. L-broth was also supplemented with the filter-sterilized solution of L-tryptophan to a final concentration of 500 µg ml−1. All inoculated flasks were incubated at 37 °C for 72 h at 120 rev. min−1. After incubation, cells were removed from stationary phase cultures by centrifugation at 2,300×g for 15 min. One ml of supernatant was mixed with 2 ml of Salkowski’s reagent. The contents in the test tube were allowed to stand in dark for half an hour for color development. The intensity of the color was measured at 535 nm by a spectrophotometer (CECIL, CE 7200).

Ultra-performance liquid chromatography

The quantification of indolic compounds in bacterial extracts was accomplished by ultra-performance liquid chromatography (UPLC). The L-broth medium was amended with 500 µg ml−1 L-tryptophan and incubated with bacterial strains at 30 °C for 72 h as mentioned above. Afterward, cells were separated from broth medium by centrifugation at 2,300×g for 15 min. Cell-free supernatant (10 ml) was further processed for the extraction and quantification of indolic compounds as described earlier (Raheem et al., 2018). The UPLC system of Waters ACQUITY H-Class (Milford, Massachusetts, USA) coupled with FD-detector (λex = 280 nm, λem = 350 nm) was used for the detection of indolic compounds. The mobile phase was 18% acetonitrile prepared in 0.1% acetic acid at a flow rate of 0.3 ml min−1. Detection of indolic compounds was performed by comparing the retention time of peaks in the samples and the mixture of authentic auxins.

ACC-deaminase activity

Screening of 1-aminocyclopropane-1-carboxylate (ACC) deaminase activity was based on the method used by Li et al. (2011). Strains were cultivated in 5 ml L-broth at 37 °C for 24 h and cell cultures were harvested in microcentrifuge tubes at 8,000×g for 5 min. Cell pellets were washed with 1 ml of liquid DF medium. Afterward, cultures were re-suspended in DF medium and placed on a shaking incubator at 37 °C for 24 h. For comparison, 2 ml un-inoculated DF-ACC medium was used as control. Next, 1 ml culture was centrifuged and 100 µl supernatant was transferred to a new centrifuge tube. Then, the supernatant was diluted to 1 ml with liquid DF medium. For a 96-well PCR-plate assay, 60 µl diluted supernatant was mixed with 120 µl of ninhydrin reagent. The Chimney-top polystyrene heat resistant PCR plate (Thomas Scientific, Swedesboro, NJ, USA) was heated on a water bath for 30 min. For comparison, 10 replicates for un-inoculated controls and samples were processed. The Riemann’s Purple color of each treated and control tube indicated the ACC-deaminase activity. For optical density, 100 µl samples were transferred to the microtiter plate and the absorbance was recorded at 570 nm with the microtiter plate reader spectrophotometer (Epoch BioTek, Winooski, VT, USA).

Phosphate solubilization

Screening for phosphate solubilization was performed following the method described by Linu, Stephen & Jisha (2009). Selected strains were streaked on the Pikovskaya medium and incubated at 28 °C for 7 days.

Hydrogen cyanide

Screening for bacterial hydrogen cyanide (HCN) production was determined following Lorck (1948) and Castric (1977) with few modifications. Glycine was added to nutrient agar, sterilized by autoclaving, and poured into Petri plates. A lawn of fresh bacterial culture was made on modified agar plates. Whatman filter paper No. 1 was soaked in a solution containing 0.5% picric acid supplemented with 2% sodium carbonate. The filter paper was placed at the top of the culture plates. These plates were incubated at 30 °C for 4 days and sealed with parafilm to avoid contamination. After incubation, the development of orange to red color was recorded.

Plant materials

The wild-type Arabidopsis thaliana L. accession Col. N6000 and the mutant lines aux1, aux4, etr1, and EIN2.5 were obtained from the European Arabidopsis Stock Centre Nottingham, United Kingdom. The mutant line aux1 and aux4 were auxin resistant and EIN2.5 and etr1 were ethylene insensitive.

Screening rhizobacteria by microphenotyping system

Strips of eight PCR tubes (Frame StripTM; 4titude, Wotton, United Kingdom), supported in 96-well PCR boxes (Thermo Fisher Scientific, Waltham, MA, USA), were filled with approximately 300 µl of autoclaved B5 Medium. For the Microphenotron bioassay, protocols of Forde et al. (2013) and Burrell et al. (2017) were slightly modified to screen the effect of bacterial metabolites on plant growth. Seeds of wild-type Columbia N6000 were surface sterilized and sown onto the agar surface (6–12 seeds per tube), stratified in the dark at 3 °C for 2–4 days. Afterward, seeds were transferred to the growth room at 22 °C with a 16 h photoperiod at a light intensity of 38 µmol m−1 s−1. The PCR boxes were kept in propagators lined with a moistened absorbent paper towel to maintain humidity. After 2–3 days, the bottom of each tube 2 mm was excised using a guillotine, and the strips were transferred to 96–well microtitre plates containing 150 µl of bacterial suspension in B5 medium. The cell density of bacterial suspension was adjusted to 107 CFU ml−1. A single lane of PCR plate that contained eight tubes was used for a single strain. Similarly, a single lane of eight tubes was used for the B5 medium as control. For each strain and control, three separate strips of eight PCR tubes were kept and the experiment was repeated three times. Tubes were incubated for 2 days to allow microbial metabolites to diffuse into the root zone. For high-throughput screening, the roots were observed for 3–4 days by removing the strips individually and scoring visually for the presence or absence of the distinctive rhizobacterial elicited root phenotype. A Canon 600D camera with a 60–mm f/2.8 lens was used for lower power imaging (Canon, Woodhatch Reigate Surrey, UK). PCR strips for each strain or control were photographed in a group of four tubes i.e., in tetrads. To measure root growth rates, the position of the tip of the most advanced primary root was marked at intervals on the side of each tube, images were captured using a Canonscan 4200 flat-bed scanner and analyzed using ImageJ 1.47v software.

Bacteria-Arabidopsis pot trials

The effectiveness of the Microphenotron system was determined by performing pot trials with A. thaliana L. (wild type) under greenhouse conditions. For pot trials, an autoclaved mixture of compost, sand, and vermiculite was used in 6:1:1, respectively. Pots were filled with medium and placed in a plastic tray adjusted with 4–5 mm autoclaved water at the bottom. Seeds were sterilized with 0.1% mercuric chloride for 3 min and placed in a fridge for 2 to 3 days to break the dormancy. Three seeds per pot were sown and placed at control temperature and light intensity and duration. For each strain, three pots were placed, and the experiment was repeated three times. The temperature of the growth chamber was 24 °C and 8/16 h dark/light duration. After 3–4 days, pots were inoculated with 10 ml bacterial suspension adjusted to 107 CFU-ml−1 with a sterile disposable syringe. After flowering (15–19 days), plants were harvested to record rosette fresh weight, leaf area, number of rosette leaves, and plant fresh weight.

The efficacy of the micro-phenotyping system was also evaluated by performing pot trials with the mutant line of A. thaliana. The involvement of bacterial phytohormones in plant growth promotion was elucidated by using four different mutant lines aux1 (auxin resistant), axr4 (auxin resistant), ein2.5 (ethylene insensitive), and eir1 (ethylene sensitive). These mutant lines were sown in plastic pots filled with potting media as mentioned above.

Statistical analysis

Data for bacterial auxin production and plant growth parameters were subjected to statistical analysis by using SPSS 20 software. Data were subjected to analysis of variance (ANOVA). Finally, mean values were separated by using Duncan’s multiple range test (P ≤ 0.05). Correlation coefficients between bacterial auxin production and cell densities at different salinity levels were also calculated (P ≤ 0.05).

Results

Identification of rhizobacteria

Sequencing analysis indicated that bacterial strains S-6, S-26, and S-50 showed similarities with genus Bacillus. All the Bacillus strains were clustered together in the phylogenetic tree (Fig. 1). S-10 and S-24 showed relatedness to genus Enterobacter. Strains S-7 and S-11 recorded similarity with genus Psychrobacter. Bacterial strains S-3, S-29, and S-80 were related to the genus Prolinoborus, Moraxella, and Pseudomonas, respectively. The sequences have been submitted to GeneBank under accession numbers KJ011870 to KJ011879.

Figure 1 Phylogenetic tree showing the relationships among different bacterial strains.

Bacterial strains followed by colored boxes were used in the present study. Strains were compared with previously isolated strains from same geographical area and submitted to GenBank (Raheem & Ali, 2015; Raheem et al., 2018). The scale bar represents a mutation per nucleotide position.

Auxin production and bacterial growth

Bacterial strains recorded variable auxin production ability at different salt stresses (Figs. 2A–2D). Overall, auxin production was negatively correlated with bacterial cell densities with increasing salinity levels (r = −1**, P ≤ 0.05). Strains of B. endophyticus S-6, E. asburiae S-24, and B. thuringiensis (S-26, S-50) S-50, grew well on 1 M NaCl stress. However, increasing NaCl concentrations recorded a decrease in bacterial cell densities. The auxin production ability of these strains was also decreased with increasing NaCl levels. All strains showed a variable IAA level in the culture supernatants. For instance, E. aerogenes S-10 and B. endophyticus S-6 recorded 105 µg IAA ml−1 at 1 M NaCl (Fig. 2). Similarly, strains S-11, S-24, S24 of Psy. Alimenterius, E. asburiae and B. thuringiensis also recorded significant auxin production.

Figure 2 Auxin production and bacterial growth under salt stress in the medium (L-broth) supplemented with L-tryptophan (500 µg ml−1) and four different salt concentrations 0, 0.25, 0.5, 0.75 and 1 M NaCl salt stress.

Value ‘r’ indicates the correlation between auxin production and bacterial cell densities at different salt concentration.

Ultra-performance liquid chromatography

Ultra-performance liquid chromatography (UPLC) confirmed the production of three kinds of indolic compounds in the bacterial culture supernatants: indol-3-acetic acid (IAA), indole-3-lactic acid (ILA) and indol-3-carboxylic acid (ICA). For IAA, maximum production was shown by Psy. alimentarius S-7 (22 µg ml−1) and E. asburiae S-24 (17 µg ml−1). On the other hand, Psy. alimentarius S-11, E. aerogenes S-10 and E. asburiae S-24, respectively, produced 21, 13, and 10 µg ml−1 ICA. Similarly, for ILA, the highest production was recorded with Psy. alimentarius S-7 (8 µg ml−1), B. thuringiensis S-50 (4 µg ml−1), and Pro. fasciculus S-3 (4 µg ml−1). Figures 3A and 3B showed the production of different indolic compounds in bacterial culture extracts and its comparison with respective standard peaks.

Figure 3 UPLC Chromatogram showing the production of different indolic compounds in bacterial culture supernatant in comparison with standard (red peak).

(A) Pro. fasciculus S-3; (B) E. asburiae S-24.

Additional plant growth promoting traits

In addition to auxin production, the majority of the bacterial isolates also showed positive results for phosphate solubilization, ACC-deaminase activity, and HCN production. For example, Psy. alimentarius S-7, E. asburiae S-24, B. thuringiensis (S-26, S-50), M. pluranimalium S-29, and P. stutzeri S-80 were able to solubilize phosphate. All the strains were able to degrade ACC and were positive for HCN production. The highest ACC-deaminase activity was exhibited by B. endophyticus S-6 (241 nmol h−1), E. aerogenes S-10 (197 nmol h−1) and P. stutzeri S-80 (190 nmol h−1) (Fig. 4).

Figure 4 Box and whisker plot analysis showing the bacterial ACC-deaminase activity.

The Microphenotron bioassay with A. thaliana

The screening bioassay through the 96-well PCR plate method with wild-type A. thaliana (Col. N6000) showed an increase in root and shoot length (Fig. 5). Strains S-26 and S-50 of B. thuringiensis recorded around 2-folds increase in root growth (Fig. 5A, Fig. S1). Similarly, M. pluranimalium S-29 and P. stutzeri S-80 also recorded significant root growth response over control. All the test strains increased root proliferation of A. thaliana in comparison with control (Fig. 6). Overall, above mentioned strains also recorded consistent results for shoot length in the Microphenotron bioassay (Fig. 5B).

Figure 5 Box and whisker plot analysis of root length (A) and shoot length (B) of A. thaliana wild type N6000 by using the Microphenotron bioassays.

Figure 6 Effect of B. thuringiensis S-26 (B) on root hair development and proliferation of A. thaliana var Col. N6000 in comparison with control (A).

Pot trials with A. thaliana L. wild-type Col. N6000

For A. thaliana wild-type Col. N6000, significant increases in leaf area, number of leaves, rosette fresh weight, and plant fresh weight were observed with bacterial inoculation as compared with un-inoculated control. A significant increase in leaf area was shown by E. asburiae S-24 (38%), M. pluranimalium S-29 (49%), and B. thuringiensis S-50 (37%) (Fig. 7A), whereas the maximum number of leaves over control were recorded by P. stutzeri S-80 (52%) (Fig. 7B). Rosette fresh biomass was observed with Psy. alimentarius S-7 and Pro. fasciculus S-3 (Fig. 8A). On the other hand, statistically significant improvements for plant fresh biomass were recorded with M. pluranimalium S-29, and Psy. alimentarius S-7, over water-treated control (Fig. 8B).

Figure 7 Box and whisker plot analysis of leaf area (A) and number of leaves (B) of A. thaliana var. Col. N6000 grown under bacterial inoculations in pot trials.

Figure 8 Box and whisker plot analysis of rosette (A) and plant fresh weight (B) of A. thaliana var. Col.N6000 grown under bacterial inoculations in pot trials.

Effects of rhizobacteria on hormonal mutant lines of Arabidopsis thaliana L.

Bacterization of A. thaliana aux1 with M. pluranimalium S-29 and E. aerogenes S-10 showed 75% and 62% increase in rosette fresh biomass, over control (Table 1). For the leaf area, Psy. alimentarius S-11 (25%), P. stutzeri S-80 (24%), and B. thuringiensis S-50 (23%) showed significant improvements. Similarly, for the number of leaves, strains S-3 and S-29 of Pro. fasciculus and M. pluranimalium recorded 84%, and 46% increases, respectively. Statistically significant increases of 1.95, and 1.67 folds in plant fresh biomass were recorded with M. pluranimalium S-29, and B. thuringiensis S-50. However, Psy. alimentarius S-11, E. asburiae S-24, and B. thuringiensis S-26 did not show the statistically significant response for respective growth parameters.

Table 1 Effect of rhizobacteria on mutant lines of A. thaliana aux1 (auxin resistant).

Strains	Rosset fresh weight (g)	Leaf area (mm2)	Number of leaves/plant	Plant fresh weight (g)	
Control	0.8 (a)	32.2 (a)	13.2 (a)	1.2 (a)	
Pro. fasciculus S-3	1.01 (c)	39.5 (cd)	24.8 (d)	1.6 (b)	
B. endophyticus S-6	0.9 (ab)	37.1 (b)	16.6 (b)	1.9 (c)	
Psy. alimentarius S-7	1.3 (d)	40.1 (d)	16.8 (b)	1.8 (c)	
E. aerogenes S-10	1.01 (c)	37.1 (b)	17.6 (b)	2.2 (d)	
Psy. alimentarius S-11	0.8 (a)	38.1 (bc)	17.1 (b)	2.6 (e)	
E. asburiae S-24	0.8 (a)	32.1 (a)	16.8 (b)	2.4 (de)	
B. thuringiensis S-26	0.9 (ab)	33.1 (a)	12.6 (a)	2.5 (e)	
M. pluranimalium S-29	1.4 (d)	37.8 (b)	18.9 (c)	3.4 (g)	
B. thuringiensis S-50	0.9 (bc)	39.4 (cd)	17.5 (b)	3.2 (f)	
P. stutzeri S-80	1.01 (c)	41.8 (e)	16.8 (b)	3.06 (f)	
Note:

Mean of three experiments (27 seedlings). Significant difference between treatments is represented by different alphabets within parenthesis evaluated by using Duncan’s multiple range test (P ≤ 0.05).

B. thuringiensis (S-26, S-50), and P. stutzeri S-80 significantly enhanced rosette fresh weight in the auxin resistant mutant (Table 2). For instance, treatment with P. stutzeri S-80, and B. thuringiensis (S-26, S-50) showed maximum improvements in leaf area. Similarly, for the number of leaves, Pro. fasciculus S-3, M. pluranimalium S-29, and B. thuringiensis S-50 showed significant increases. For fresh biomass, B. thuringiensis S-50 (189%), P. stutzeri S-80 (162%), and E. aerogenes S-10 (86%) were the most promising. However, axr4 treatment with E. asburiae S-24 did not record any response for rosette fresh weight or leaf area, over control.

Table 2 Effect of rhizobacteria on mutant lines of A. thaliana axr4 (auxin resistant).

Strains	Rosette fresh weight (g)	Leaf area (mm2)	Number of leaves/plant	Plant fresh weight (g)	
Control	0.63 (a)	32.8 (b)	13.5 (a)	1.4 (a)	
Pro. fasciculus S-3	0.91 (bc)	35.8 (c)	25.8 (f)	2.6 (b)	
B. endophyticus S-6	0.66 (a)	36.5 (cd)	17.8 (bc)	2.4 (b)	
Psy. alimentarius S-7	0.93 (c)	39.5 (f)	18.6 (cd)	2.4 (b)	
E. aerogenes S-10	0.86 (bc)	37.1 (cde)	19.5 (de)	2.7 (b)	
Psy. alimentarius S-11	0.90 (bc)	37.5 (de)	18.7 (cd)	2.7 (b)	
E. asburiae S-24	0.75 (ab)	31.3 (a)	17.8 (bc)	2.4 (b)	
B. thuringiensis S-26	0.97 (c)	39.9 (f)	13.8 (a)	2.6 (b)	
M. pluranimalium S-29	1.4 (e)	38.2 (e)	19.9 (e)	3.5 (c)	
B. thuringiensis S-50	1.4 (e)	40.1 (f)	18.9 (d)	4.2 (d)	
P. stutzeri S-80	1.2 (d)	42.1 (g)	17.5 (b)	3.8 (c)	
Note:

Mean of three experiments (27 seedlings). Significant difference between treatments is represented by different alphabets within parenthesis evaluated by using Duncan’s multiple range test (P ≤ 0.05).

Inoculation of ein2.5 with M. pluranimalium S-29, P. stutzeri S-80, and B. thuringiensis S-50 showed up to one-fold increase for rosette fresh biomass, over control (Table 3). Similarly, the leaf area was enhanced with B. thuringiensis S-26, E. aerogenes S-10, and M. pluranimalium S-29. A statistically significant increase in the number of leaves was recorded with M. pluranimalium S-29 (49%), B. thuringiensis S-26 (34%), and P. stutzeri S-80 (23%). For fresh biomass, maximum improvements were recorded with B. thuringiensis S-50, P. stutzeri S-80, and Psy. alimentarius S-11, respectively when compared with un-inoculated control. However, a statistically insignificant response was recorded for rosette fresh weight with Psy. alimentarius S-7, Psy. alimentarius S-11, and E. asburiae S-24.

Table 3 Effect of rhizobacteria on mutant lines of A. thaliana ein2.5 (ethylene insensitive).

Strains	Rosette fresh weight (g)	Leaf area (mm2)	Number of leaves/plant	Plant fresh weight (g)	
Control	0.7 (c)	33.7 (a)	14.3 (d)	1.9 (abc)	
Pro. fasciculus S-3	0.7 (b)	33.5 (a)	12.8 (b)	1.6 (a)	
B. endophyticus S-6	0.6 (a)	34.5 (a)	13.8 (c)	1.8 (ab)	
Psy. alimentarius S-7	0.8 (c)	33.9 (a)	12.3 (a)	2.09 (bc)	
E. aerogenes S-10	0.8 (c)	33.6 (a)	15.1 (e)	2.2 (c)	
Psy. alimentarius S-11	0.8 (c)	36.4 (b)	15.8 (f)	2.5 (d)	
E. asburiae S-24	0.6 (ab)	41.7 (d)	12.6 (ab)	1.8 (a)	
B. thuringiensis S-26	1.6 (e)	45.1 (e)	19.2 (h)	2.7 (d)	
M. pluranimalium S-29	1.2 (d)	40.1 (c)	21.4 (i)	3.6 (e)	
B. thuringiensis S-50	1.6 (e)	41.1 (d)	15.3 (e)	4.5 (f)	
P. stutzeri S-80	1.6 (e)	36.2 (b)	17.6 (g)	3.8 (e)	
Note:

Mean of three experiments (27 seedlings). Significant difference between treatments is represented by different alphabets within parenthesis evaluated by using Duncan’s multiple range test (P ≤ 0.05).

Bacterization of etr1 stimulated rosette growth with P. stutzeri S-80, B. thuringiensis S-50, and Psy. alimentarius S-11, over control (Table 4). For leaf area, P. stutzeri S-80, B. thuringiensis S-50, and Psy. alimentarius S-11 recorded maximum improvements. For the number of leaves, Pro. fasciculus S-3, E. aerogenes S-10 and M. pluranimalium S-29 were the most effective. In the case of plant fresh biomass, B. thuringiensis (S-26, S-50), and P. stutzeri S-80 showed up to 1.9 fold increases over respective un-inoculated controls. However, in case of rosette or plant fresh weight, a statistically insignificant response was recorded with Pro. fasciculus S-3, B. endophyticus S-6, Psy. alimentarius S-7, E. aerogenes S-10, and B. thuringiensis S-26, over untreated control.

Table 4 Effect of rhizobacteria on mutant lines of A. thaliana etr1 (ethylene insensitive).

Strains	Rosette fresh weight (g)	Leaf area (mm2)	Number of leaves/plant	Plant fresh weight (g)	
Control	0.8 (a)	33.5 (a)	14.1 (a)	1.6 (a)	
Pro. fasciculus S-3	0.9 (a)	37.3 (c)	24.9 (d)	1.6 (a)	
B. endophyticus S-6	0.7 (a)	35.9 (bc)	18.9 (bc)	1.9 (ab)	
Psy. alimentarius S-7	0.9 (a)	40.3 (d)	18.1 (b)	1.9 (ab)	
E. aerogenes S-10	0.8 (a)	35.2 (b)	19.6 (c)	1.9 (ab)	
Psy. alimentarius S-11	1.3 (b)	36.9 (c)	19.2 (bc)	2.2 (b)	
E. asburiae S-24	0.9 (a)	32.4 (a)	19.4 (c)	2.2 (b)	
B. thuringiensis S-26	0.8 (a)	39.7 (d)	18.7 (bc)	3.4 (c)	
M. pluranimalium S-29	1.5 (c)	41 (de)	19.4 (c)	4.03 (d)	
B. thuringiensis S-50	1.6 (c)	41.9 (ef)	19.2 (bc)	4.7 (e)	
P. stutzeri S-80	1.7 (c)	42.6 (f)	18.9 (bc)	4.01 (d)	
Note:

Mean of three experiments (27 seedlings). Significant difference between treatments is represented by different alphabets within parenthesis evaluated by using Duncan’s multiple range test (P ≤ 0.05).

Discussion

The Microphenotron is an automated microphenotyping platform that can be used to study plant growth and development under controlled conditions. By using this approach, the phenotypic effect of a large number of small molecules can be assessed. Initially, this system was developed to test the effect of synthetic chemicals on root architecture (Forde et al., 2013). This system may have the potential to screen beneficial PGPR by using A. thaliana. This screening system may demonstrate the effect of secondary metabolites produced by PGPR on plant growth. In the present study, A. thaliana was used to further study physiological responses by which rhizobacteria influence plant growth and development. Overall, auxin-producing bacterial strains promoted the growth of Arabidopsis in Microphenotron and pot trials (Fig. 9). In previous studies, it was reported that various PGPR can promote plant growth by using complex hormonal signaling (López-Bucio et al., 2007; Pieterse et al., 2012). From these experiments, we can suggest that the effects were mediated by phytohormones signaling (Mayak, Tirosh & Glick, 1999; Belimov et al., 2005; Chen et al., 2013).

Figure 9 Biplot analysis to compare the effect of bacterial inoculations on root length and rosette fresh weight of A. thaliana var. Col. N6000 in Microphenotron and pot trials, respectively.

Forde et al. (2013) developed a 96-wells PCR plate method to screen the antagonistic and agnostic effect of different amino acids on the root architecture system. In this study, PGPRs were screened in the Microphenotron to evaluate their effect on the growth of Arabidopsis and their root architecture system (Fig. S1). Results demonstrated that the majority of the PGPRs can trigger root proliferation in inoculated plants. For instance, the highest increase of 2-fold was recorded with B. thuringiensis (S-26, S-50) (Fig. 5A). Similarly, statistically significant growth enhancement for different vegetative parameters was also recorded in pot trials by using A. thaliana wild-type Col. N6000 (Figs. 7 and 8, Fig. S2). For pot trials, significant growth responses were recorded with M. pluranimalium S-29 (leaf area), Psy. Alimentarius S-7 (Rosette fresh weight), P. stutzeri S-80 (number of leaves) and M. pluranimalium S-29 (plant fresh weight). Overall, tested strains showed consistent results in Microphenotron and pot trials (Figs. 5, 7–9). For instance, B. thuringiensis (S-26, S-50), M. pluranimalium S-29 and P. stutzeri S-80 recorded growth improvements for root length and rosette fresh weight in both Microphenotron and in pot trials, respectively. PGPR has been shown to stimulate biomass production, primary and lateral root development in A. thaliana (Gutiérrez-Luna et al., 2010).

To investigate the role of bacterial auxin and ethylene signaling, mutant lines of A. thaliana impaired in auxin or ethylene signaling were used in pot rials under greenhouse conditions. These mutant lines included aux1 (auxin resistant), axr4 (auxin resistant), ein2.5 (ethylene insensitive) and etr1 (ethylene insensitive). Pot trials with A. thaliana (N6000) showed significant enhancement in growth parameters with Psy. alimentarius S-7, M. pluranimalium S-29, and E. asburiae S-24, respectively. Experiments with mutant lines impaired for auxin signaling showed that B. endophyticus S-6, Psy. alimentarius S-11, E. asburiae S-24, and B. thuringiensis S-26 recorded comparable results to that of control. Similarly, for ethylene insensitive mutant lines (ein2.5 and etr1), growth responses with Pro. fasciculus S-3, B. endophyticus S-6, Psy. alimentarius S-7, E. asburiae S-24, and B. thuringiensis S-26 were statistically comparable with control. It can be suggested that auxin or ethylene signaling may be involved in plant growth promotion by these PGPR. However, the majority of the PGPR also enhanced plant growth of auxin or ethylene resistant mutant lines. López-Bucio et al. (2007) also demonstrated that plant growth-promoting attributes of rhizobacteria may work independently of ethylene or auxin signaling. It may suggest the involvement of other plant growth-promoting attributes that might play a role to mediate the growth of Arabidopsis. According to similar studies with fungus, growth of etr1, ein2, and ein3/eil1 mutant was not promoted or/and even inhibited by the inoculation (Camehl et al., 2010).

Conclusions

In conclusion, the Microphenotron based on the 96-wells PCR plate method was very effective for screening PGPR for agricultural applications. This in vitro bioassay can readily demonstrate the effect of bacterial metabolites on the rooting of A. thaliana. Based on the Microphenotron growth response, PGPR may be further selected for plant experiments under greenhouse or natural environmental conditions. This study further suggested that plant growth promotion by rhizobacteria may be mediated by a variety of mechanisms. For a few strains such as Psy. alimentarius S-11, E. asburiae S-24, and B. thuringiensis S-26, auxin or ethylene signaling may be involved in plant growth and development. In addition to that, other mechanisms like ACC-deaminase activity, phosphate solubilization, or HCN production may also be playing a critical role.

Supplemental Information

Supplemental Information 1 Data for statistical analysis for Fig. 2.

Click here for additional data file.

Supplemental Information 2 Effect of bacterial inoculation on vegetative growth parameter of A. thaliana var. Col.N6000.

Click here for additional data file.

Supplemental Information 3 Screening bioassay through novel 96 well PCR plate method.

Tube on the left side is control followed by Pro. fasciculus S-3, B. thuringiensis S-26, M. pluranimalium S-29 and P. stutzeri S-80 under controlled axenic condition.

Click here for additional data file.

Supplemental Information 4 Pot trials with A. thaliana wild type (Columbia N6000), inoculated with B. thuringiensis S-26, M. pluranimalium S-29 and B. thuringiensis S-50 in comparison with un-inoculated control.

Click here for additional data file.

I would like to express my sincere gratitude to Dr. Ian Dodd, and Brian G. Forde, Lancaster Environment Center, University of Lancaster, United Kingdom, for the continuous guidance and provision of the research facility, for their patience, motivation, enthusiasm, and vast knowledge.

Additional Information and Declarations

Competing Interests

Author Contributions

DNA Deposition

Data Availability

The authors declare that they have no competing interests.

Asif Raheem performed the experiments, prepared figures and/or tables, and approved the final draft.

Basharat Ali conceived and designed the experiments, analyzed the data, authored or reviewed drafts of the paper, and approved the final draft.

The following information was supplied regarding the deposition of DNA sequences:

The sequences are available at GenBank: KJ011870 to KJ011879.

The following information was supplied regarding data availability:

The raw data are available as Supplemental Files. The sequences are available at GenBank: KJ011870 to KJ011879.

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
