# Peer review of "The Microphenotron: a novel method for screening plant growth-promoting rhizobacteria"

_PeerJ, doi:10.7717/peerj.13438_

## Round 0.1 · original submission · Major Revisions

Your manuscript was reviewed by three experts with firsthand knowledge of these systems. I consider all three reviews to be top quality, and I expect the authors to provide a thorough and attentive response to each of the points raised by all three reviewers. I must emphasize that the reviews were somewhat critical, and it is my opinion that this manuscript is "teetering on the edge" of rejection by the journal because of a lack of attention paid to properly justifying the replication and properly specifying the controls in the experiments. If the rebuttal attends carefully and thoroughly to these matters, demonstrating that the replication is sufficient and appropriate and detailing how each of the controls is appropriate and where additional controls were lacking (and why) then the manuscript can be considered further. Professional editing by a native English speaker or commercial service is called for.

Reviewer 1 ·

Basic reporting

1)The text requires grammatical polishing. In its current station it is difficult to understand.
2)The images presented have extremely low resolution. I would encourage the submission of high quality figures. The narrative is quite simple so it would benefit from having decent resolution figures.
3) I think the data presented in the Tables should be transformed into figures. I find it quite weird that those experiments are summarized as tables and not as figures. This is particularly relevant for the tables in which the auxin and ethylene lines were tested.
4) I could not see any reference to raw data. Please provide it

Experimental design

1)The introduction requires formatting. The research question is well defined but still it is difficult to grasp from the introduction. The question that the authors formulate is valid despite lacking novelty.
2) The clear relevance of the paper is that the authors implement a novel protocol to maximize the throughput of PGP traits.
3)The methodology is clearly stated

Validity of the findings

1)Authors do not mention how many times each experiment was performed nor mention how reproducible their 96 well plate assay is. I am curious to see how reproducible are the measurements they perform. Due to the fact that the major novelty of this article is this trough-put screening I would encourage that they asses how reproducible and reliable their measurements are. This should be added to the article.
2) The authors mention a validation of the approach using a green house experiment. Nevertheless, despite mentioning this they never actually compare the trends observed in their high trough-put system in comparison to the green house experiment. I want to see a correlation performed between both approaches, if their high trough-put assay is proper enough we should expect to see high correlation between the results obtained by both approaches. Specifically, the author should correlate a specific trait (e.g, fresh weight) between both approaches and show it quantitatively. Due to the fact that their paper is based on this methodological novelty it is quite relevant that they show its validity.

Reviewer 2 ·

Basic reporting

Manuscript is generally well written but does need some English editing. Figures and tables need to be improved significanly and the methods need more details, specifically about replication. The table say that the values are means of 3 reps which is severly insufficient.

Experimental design

There are problems here, as I point out in my general comments. Some controlls are missng in my opinion, replication is not sufficient, and some experiments need to be preformed side by side.

Validity of the findings

no comment

Additional comments

In this manuscript, the authors screen 10 bacterial isolates for plant growth promoting traits.
These isolates are screened in vitro for a number of traits associated with plant growth promotion (more on that later), then screen for their activity in a lab-based plant growth phenotyping system, and finally validate their findings in soil. The authors also include the use of phytohormone insensitive and hypersensitive Arabidopsis mutants, to test whether hormonal manipulation indeed plays a role as a pgpr mechanism.
This is a widely used experimental pipeline, with the main point of novelty being the use of the “micro-phenotron” system. It seems like an intriguing system, but its main advantage of allowing high throughput screening is not demonstrated here. Only 10 bacterial strains are tested with 3 only 3 replicates, which is not sufficient power to deduce anyting about plant growth promotion. Furthermore, this paper does not address the two underlying assumptions: (1) that the in vitro plant growth promotion screens are predictive of activity in plants and (2) that the phenotron results correlate with results in soil. This is especially acute in the case of auxin production, which is many cases is actually harmful for the plant. Curiously, all the tested strains seem to possess the growth promotion traits screened for and all seem to produce larger plants than the uninoculated control (critically – it is not clear whether the phenotron remains axenic or not), so there is no true negative control of plants inoculated with non pgpr strains.
The use of hormone insensitive mutants is interesting, and it seems to support the notion that the growth promotion observed is not phytohormone related, but these experiments need to be performed with wt and the mutants side by side in order to be informative, and this does not seem to be the case.
Finally, the data presentation in this manuscript needs to be improved, as does the number of replicates. I recommend presenting the data using boxplots with scattered dots to show the true distribution of the data.

Specific comments.

The introduction focuses on bacterial IAA production as a reliable measure for plant growth promoter potential. I do not agree with this premise. IAA production can actually be harmful for the plant, and the evidence cited by the authors to support their claim is not sufficient.

Line 234 It’s not entirely clear how taxonomy was assigned to the isolates. What database were they compared to and in what method. How similar are they to the nearest hit?
In addition, the tree is not informative if it only includes your own isolates. You could make a combined tree with related database sequences to inform the reader about the relatedness of your isolates to known sequences. Furthermore, it is not clear how you have 2 isolates on the tree with seemingly identical sequences but different genus names assigned (Psychrobacter and Moraxella).

Line 245 There is too much text here to explain something trivial. Just say that reduction in auxin production with increasing salinity is explained by reduced growth. Figure 2 is barely readable.

Line 268 If I understand correctly, all isolates tested were positive for all pgpr traits tested? I find this 100% success rate puzzling. Also, the lack of negative strains does not help in assessing the predictability of these in vitro screens for actual plant growth promotion, which is an ongoing lacuna in the field.

Line 277 Make sure all figures are called in the text. Figure 5 needs to be improved. Present it as a box plot showing all datapoints. Again, it is puzzling that all strains were positive for growth promotion. This makes me wonder whether this is an artifact of the system. Would E. coli also produce growth promotion in this system? A proper negative control is needed (e.g bacterial stains with no growth promotion activity.

·

Basic reporting

Potted plant experiments are costly, difficult to reproduce and often don’t translate to the field. To address this Forde’s group at Lancaster University developed a “microphenotron” that uses a model plant grown in modified 96 well plates. As I understand it the microphenotron, was developed for screening metabolites. The application of this technology to screening rhizobacteria is interesting but I am not sure if the authors clearly evaluated, or at least presented, this application of the microphenotron. Discussion includes a statement that “system can be used to screen beneficial rhizobacteria” but that statement is not supported by data or a reference (line 336).

Introduction should be modified to state if and when the microphenotron has been applied to screening live bacteria, versus metabolites and small molecules. This should be done in a dedicated paragraph that justifies using this technology. The last paragraph of Introduction should state what was done and what was observed. Results should include a comparison of the results obtained with the microphenotron and potted plants and Discussion should state whether the system worked in this application.

Line 26. Start Abstract and Introduction with background on why the microphenotron is used.
Line 29. Replace “would allow” with “allows”
Line 30. Replace “were able to degrade” with “degraded”
Line 58. It is not clear here if you are talking about the model plant (Arabidopsis) or the microphenotron.
Line 72. Introduction needs reorganization. Replace “species” with “strains,” delete “have been reported to” and move that and line 72-75 to the start of the 1st paragraph, where you introduce the field of PGPR.
Line 76. Here and throughout, delete fillers like “Now it is established through numerous studies...” and “It was established from different studies (Line 347). Also, replace “have been considered as” with “is”, unless you have doubts about that science.
Line 89 – 99. This section needs revision, following the above general comments. Delete “that would” and “under specified..effect.”
Line 93. Here and throughout, the name is Forde not Ford.
Line 96-99. This discussion of root morphology doesn’t belong here.
Line 104. Provide location of sampling areas and mode of collection.
Line 135. Why cap Indole?
Line 148. Provide (City, State) for USA locations.
Line 169. Why in the plate reader identified here but not in line 133.
Line 177. Sterilized.
Line 210. Revise to 6:1:1
Line 235 and 240. Move “Sequences were... BLAST” and “The sequences have...KJ011879.” to Methods.
Line 246. Start each paragraph in Results with a topic sentence that states the key result. Lines 246 – 248 provide no results. Similarly, move Line 269 -271 to Methods.
Line 282. Tell the reader what the figure shows in terms of the treatment effect.
Line 322. Replace “showed stimulatory effects on rosette growth biomass” with “stimulated rosette growth”
Line 328. Change to “controls”
Line 363. Delete “parameters”
Figure 5. Box, whisker, violin or similar plots are preferred to bar charts.

Experimental design

What is an appropriate control for this application? The control is not identified in Table 2 or Methods, so I presume it is water. I am not sure if that is an appropriate control for tests of live bacteria. All of the isolates significantly increased root growth relative to water, so the system might reveal traits not associated specifically with PGPR. In other words, a more appropriate control might be a strain that does not promote plant growth. This would control for the fertilization effect of adding microbial biomass.

Validity of the findings

I am concerned that the each 8-well strips used in the system corresponded to one treatment or the control. Some might argue that each strip should be treated as one replicate, as the treatments are not randomly assigned. I would be interested in the strip to strip variation for duplicate treatments.

---

## Round 0.2 · Major Revisions

All three reviewers reviewed your revised manuscript, but many were left dissatisfied. Reviewer 3 felt that you glossed over many of their recommendations in your responses, and I concur.

Where I am most hesitant in considering your manuscript for acceptance is my agreement with the lack of controls as outlined by Reviewer 2. As the reviewer states, without the appropriate controls, the experiments do not address the questions posed: without strains that either explicitly lack PGPR traits or a random selection of soil strains as controls you cannot present this as a controlled study and cannot test hypotheses as stated.

However, as Reviewer 3 (and Reviewer 1) point out, there is value in publishing your data and your manuscript, as it does represent solid data on the application of this system to field and greenhouse studies.

Here is my stern guidance for me to consider acceptance:

1) Thoroughly revise your Title, Abstract, Introduction, and Discussion to clarify that you cannot test hypotheses related to true screening with this design because you don't have proper controls. As such the method is good for characterizing and perhaps selecting isolates but you need to write a strong justification of how this method "screens" for PGPR traits.

2) As R1 and R3 note, figures remain illegible and the structure of the manuscript is still weak. Please follow R3's recommendations on restructuring and do a thorough rewrite. Simplify the abstract to avoid excessive technical details - this is a summary, not a results section.

3) Write a thorough response to all reviewer points, paying special attention to how you address the very important criticisms leveled by Reviewer 2, which I tend to agree with.

Thank you.

Reviewer 1 ·

Basic reporting

I appreciate the inclusion of the phylogenetic tree of isolates assayed in this study.
Some legends in the figures are not legible still.

Experimental design

Ok

Validity of the findings

The authors lacked the clear correlation against a proven system.

Reviewer 2 ·

Basic reporting

no comment

Experimental design

no comment

Validity of the findings

no comment

Additional comments

- My concerns regarding sample sizes and phylogenetic placement were addressed. On the other issues, I stand by my original assessment. I think the Microphenotron system can be a great system for screening PGPRs. I also think that using hormone-insensitive mutants is an elegant way to test whether PGPR in these strains is hormone-dependent. However, the experiments as they are presented don’t really address the questions that the authors set out to answer. The authors say that these 10 strains were selected since they were found to have many PGPR traits a-priori. I would like to know what happens in their system with strains that were not found to have these traits. In other words, how does PGPR in their system correlate with the results of their pre screening on one hand, and to soil based assays on the other. Since all strains in this study seemed to have a positive effect, the only negative control is uninoculated plants and I do not think that is sufficient. Why weren’t strains that were NOT shortlisted, meaning that they didn’t perform as well in the in-vitro assays, used as a negative control in the in-planta assays?

The authors and I are in some disagreement as to the role of rhizobacterial auxin
production as a plant growth promoting mechanism. I think this is a legitimate debate, and certainly not ground for rejection. However, they do not really provide literary support for their claim. In line 90 they write: “For agricultural applications, screening of rhizobacteria based on auxin production has provided a reliable tool for the selection of PGPR” and provide Pavlovic, 2018 as a reference for this statement. The paper that is referred to did not study PGPRs at all.

Finally, I suggested that the authors use boxplots and indeed they Figure 7 was added with boxplots in it. But this figure presents what by all accounts is a meaningless comparison.

I am sorry I cannot be more supportive, but I do not think that this manuscript should be published as it is presented. It seems like a more methodical comparison of their system could be made with both pot-based assays and PGPR trait screenings.

·

Basic reporting

The revised manuscript presents a very important data set. As far as I know, this is the first study to apply the microphenotron to the screening of rhizobacteria for the ability to promote plant growth. The parallel measurement of plant growth promotion in potted plant trials, which is current standard, adds particular power to this work. As in the first version, the authors need to improve the organization of the manuscript to present this key contribution clearly.

Experimental design

The research question needs to be clearly defined. The manuscript does not make it clear that this is a novel application of the microphenotron.

Validity of the findings

Results should have a paragraph that starts with a topic sentence that states something like what the rebuttal says “Strains that performed well in ‘Microphenotron’ bioassay also showed good results in pot trials for wild-type A. thaliana.” That paragraph should provide a figure that compares the microphenotron to potted plant trials. The x-axis should present plant growth response in the microphenotron (with errors bars that indicate the variation between three different experiments) and the y-axis presents the result of the potted plant trials (with error bars for within treatment variation). Each point should correspond to an individual isolate.

Additional comments

As many of my previous comments were ignored or glossed over, I am not going took a fresh look at the manuscript.
Reorganize Introduction into the following paragraphs.
1. Provide background why we need more PGPR and the limitations of the current methods (in vitro traits and potted plant trials) of screening for them. Move lines 75 – 97 to this first paragraph.
2. Start this paragraph with a topic sentence that describes the microphenotron. Provide a topic sentence that clearly sone way or another if this system has been evaluated for screening bacterial isolates for plant growth promotion. In other words, is this work novel?
3. State what you did: Rhizobacteria were isolated from halotolerant plants because ___. Strains tested with in vitro assays for traits associated with PGPR, potted plant trials and a microphenotron. Salt concentrations was varied in in vitro bioassays because _____ and different KO mutants of Arabidopsis were selected because _____. Then summarize the results, the microphenotron agreed/disagreed with potted plant trials.

Specific comments.
Line 27. Replace “deliver…” with “test the response of seedlings of Arabidopsis to natural products”
Line 28. Delete “of shoot…where”
Line 29. Delete “can be assessed”
Line 31. You cannot assume they are PGPR. Replace with “rhizobacteria”
Line 32. Add comma before “which”
Line 38. Revise to “extracts, including..”
Line 39. Here and through write actively by starting the sentence with the subject (rhizobacteria) and the verb what the subject does (produced ___). I suggest “Psycrobacter ___ and ____ produced the highest amount of IAA….” Leave the amount they produced to Results.
Line 50. Which results show that microphenotrons are effective? The current standard (potted plant trials), which was used to validate the results, is not mentioned in Abstract. As I understand it the “experiments with mutant lines” were potted plant trials designed to elucidate the mechanism of plant growth promotion for these isolates but potted plant trials with the wild type was used to validate the microphenotron.
Line 75-76. Comma before “such” and replace “or” with “and.”
Line 77-79. Delete “After screening…phytohormones but”
Line 81-85. Why go into so much about detail about one trait (auxin) and not the others (HCN, P..). The discussion of tryptophan and auxin is distracting. Delete it.
Line 176. Delete “the method suggested by”
Line 238. As stated in my earlier review, move “Sequences were... BLAST” and “The sequences have...KJ011879.” to Methods. Start each paragraph in Results with a topic sentence that states the key result.
Line 241. The term “similarly” and the phrase “On the other hand” add nothing.
Line 290. As stated in my earlier review, tell the reader what the figure shows then cite the figure.
Line 293-295. Move this to Methods and/or Introduction. Also, make it clear in the Tables how the data was generated (microphenotron or potted plant).
Line 304. Revise to write directly “Application of B. thuring… significantly enhanced rosette…in the auxin resistant mutant.”
Line 321. Delete “showed” here and in line 323.
Line 339. Delete “were found to have the ability to” and change to “promoted”
Line 351. Delete phrases like “It was established from different studies that,” which have no content. State the fact and cite the source.
Line 462. Carefully check references for content and consistency of format. For example, Martinez (2020) lacks a year and some references provide the issue (line 458: 86(1):1-..” and some do not (line 466: 2:49-…).
As stated in review 0, “Figure 5. Box, whisker, violin or similar plots are preferred to bar charts.”

---

## Author Rebuttal · Round 0.2

With reference to article #52899 titled "Microphenotyping system: A novel method for screening plant growth-promoting rhizobacteria" a revised version of the article is submitted for further consideration and processing. Article has been revised as per Editors and worthy reviewers' suggestions. Point to point response follow below:

## RESPONSE TO EDITORS COMMENTS

We are very thankful to the Editor to provide us with an opportunity to revise this article. All the sections of this article have been revised keeping in view the reviewers' comments. In methodology, number of replicates for each experiment are now mentioned with clarity in the revised version. Different sections of the methodology have been supported with relevant references. Figures number 1-7 have been modified to increase the resolution as per Journal's requirement. Moreover, effort has been made to remove the grammatical errors in the article.

## RESPONSE TO "REVIEWER 1" COMMENTS

*Basic reporting*

1). Effort has been made to improve the English text or to remove grammatical errors in the revised version of this article.

2). The resolution of submitted figures was compromised during the submission process and their conversion to PDF file. Good quality figures are now provided in the revised manuscript as per Journal's requirement.

3). Number of tables have been reduced to 4. Table No. 1 and 2 has been converted to figures. However, to elaborate the growth response of different treatments or growth parameters of mutant lines of *Arabidopsis thaliana* data is presented in the form of tables.

4). Reference to raw or supplemental data is now given in the revised article as per worthy reviewer's suggestion.

*Experimental design*

1). The whole text of introduction has been formatted and research question is now well defined by adding more literature to justify the use of 'Microphenotron' for screening plant growth-promoting rhizobacteria (PGPR).

2). We are thankful for" reviewer 1" comments that in current study we are suggesting novel protocol based on 'Microphenotron' to screen bacteria for PGP traits.

3). We are thankful for reviewer's comment that methodology is clearly stated.

*Validity and findings*

1). Number of replications for each experiment is now elaborated in the revised version of the article. For instance, for 'Microphenotron' for each strain or control, 8 wells or PCR tubes were used in triplicate and experiment was repeated three times. Overall, 72 wells or tubes were used for each strain to authenticate the reproducibility of the experiment. This revision had been mentioned on page 8 lines 212 to 214. Similarly, for bacteria-*Arabidopsis* pot trials, number of replicates has been mentioned in lines 230-232.

2). The trend of growth response for 'Microphenotron' and green house experiment is mentioned in the discussion section. In 'Microphenotron' screening was based on root or shoot growth response. Such parameters were not mentioned in pot trials with wild-type Col. N6000. Therefore, we did not prefer to perform correlation analysis between unrelated parameters such as root growth and rosette fresh weight or number of leaves. I hope worthy reviewer may agree with this point.

**RESPONSE TO "REVIEWER 2" COMMENTS**

*Basic reporting*

1). We are very thankful for the worthy reviewer's comments that this manuscript is generally well written. All the major sections of this article have been revised according to reviewer's suggestions. Grammatical errors have been removed and different sections have been rephrased. Especially, reviewer's concern about the replication of different experiments has been properly addressed. In first draft of the article, mean of three replicates was mentioned with reference to three different set of experiments. In tables and in text this confusion has been removed in the revised article. Figures may have lost resolution during the formation of PDF file. However, modified figures with high resolution are submitted with the revised manuscript.

*Experimental design*

1). For bacterial biochemical screening and for pot trials, number of replications and respective control treatments are now mentioned clearly in different sections of the methodology.

*Response to general comments*

1). In present study, 10 bacterial strains were short listed based on their multiple plant growth-promoting traits, especially on auxin production. Initially, we isolated number of bacterial strains that are now indicated in figure 1 and supported with reference (Raheem et al., 2018). Finally, 10 bacterial strains were selected for further experimentation for this study. Bacterial strains were

evaluated by performing different set of experiments with *Arabidopsis* including phytohormone mutant lines.

2). Reviewer's concern with reference to number of replicates is now clarified in the revised article. For 'Microphenotron' bioassay single lane of eight PCR tubes or wells was placed for each strain or control in three replication and experiment was repeated three times. Overall, 72 PCR tubes were used for single strain or B5 media (control). Similarly, for pot trials experiment was also repeated three times with overall 27 seedlings inoculated for each strains or control. Although, more replicates may be included to increase the authenticity of the experiments. But overall, currently used number of replicates may be sufficient to predict the outcome of the results. Moreover, our results indicated that strains that performed well in 'Microphenotron' bioassay also showed good results in pot trials for wild-type *A. thaliana.* Auxin production by PGPR strains in soil is not always detrimental to host plant. Nevertheless, reviewer's concern may be right for some plant pathogenic bacteria that may induce abnormal growth in plants due to increased auxin production. But for free living PGPR, auxin production is very beneficial as supported with number of references in introduction (Egamberdieva, 2012; Iqbal and Hasnain, 2013; Aslam and Ali, 2018; Raheem et al., 2018). In present study, all the strains possess beneficial traits as they were selected from several bacterial isolates as mentioned in figure 1. For 'Microphenotron' bioassay, B5 medium was used as a control in comparison with bacterial inoculations. For negative control, bacterial strain was not used as we do not have culture of bacteria that may be deficient for number of plant growth promoting traits.

3). In this article, wild-type and mutant lines of *A. thaliana* were already compared in different set of experiments in 'Microphenotron' and then in pot trials.

4). Two figures have been added in the manuscript to improve the presentation of data. Moreover, figure 7 has been converted into Box and Whisker plot to analyze the plant growth response for different growth parameters.

***Response to specific comments***

1). Bacterial auxin production was used as a major criterion to screen PGPR. Reviewer's concern about the detrimental effects of auxin may be true for some plant pathogenic bacteria as mentioned above. However, free living PGPR produce optimum level of auxin within the rhizosphere due to the limited availability of L-tryptophan in root exudates. In introduction section, we have quoted a few references that showed beneficial effects of microbial auxin.

2). Line 234: Bacterial strains were identified by comparing already submitted sequences in GenBank through Blast analysis. Their similarity is now elaborated in lines 249-255 in the revised manuscript.

3). Phylogenetic tree is modified to include all possible sequences (25 strains) from the same geographical location. Phylogeny is assigned according to nearest related sequence. Moreover, *Psychrobacter* and *Moraxella* are now separated and occupying different branches on the phylogenetic tree.

4). Line 245: Sentence has been modified and rephrased as per reviewer's suggestion. Correlation between bacterial auxin production and cell densities is clarifies in lines 259-261 in the revised version. Figure 2 has been modified to increase the resolution and submitted with the revised manuscript.

5). Line 268: Initially, we isolated number of bacterial strains that is mentioned in the legend of figure 1. Next, we selected only those strains that were positive for multiple plant growth promoting traits. That is why all 10 strains were positive for different traits. Negative strains were not used as we do not have bacterial isolates that are deficient for all beneficial attributes.

6). Line 277: All figures are now included in the text including the supplemental files. Figures with high resolution have been provided with the revised manuscript. As mentioned above, we selected only 10 strains with positive plant growth promoting traits from several bacterial isolates. Moreover, *E. coli* has not been used in present study as PGPR. We used two strains of *Enterobacter aerogenes* that were reported in number of studies/ in literature as PGPR.

**RESPONSE TO "REVIEWER 3" COMMENTS**

We are very thankful for the valuable comments of worthy reviewer to improve the text of this article. Point to point response to comments is given below.

***Basic reporting***

1). PGPR may show variable interaction with inoculated plants that may depend upon bacterial species or plant used in the experiment. Sometime, PGPR may show inconsistencies in results but in majority of the cases inoculated plants recorded promising growth response. In present study, we have used 'Microphenotron' to screen our potential PGPR strains for further application. Moreover, line 336 has been modified to remove the confusion for the reader.

2). Introduction has been modified as per reviewer's suggestion. The use of 'Microphenotron' has been justified in a dedicated paragraph and supported with relevant references. Results of 'Microphenotron' and pot experiments have been compared in the discussion section.

3). Line 26: Abstract and introduction has been started with the background of 'Microphenotron'. Line 26-31 have been modified in abstract and lines 56-64 have been incorporated in introduction as suggestion by the reviewer.

4). Line 29: "would allow" replaced with "allows".

5). Line 30: "were able to degrade" has been replaced with "degraded."

6). Line 58: Here we are talking about the application of 'Microphenotron' by using *Arabidopsis thaliana* as a model system. Now sentences have been modified according to reviewer's comments.

7). Line 72: Major sections of the introduction has been reorganized. Moreover, "species" has been replaced with "strains". Similarly, "have been reported to" is deleted in the revised article and Lines 72-75 have been moved at the start of the first paragraph.

8). Line 76: Fillers have been deleted as per reviewer's suggestion. Also "have been considered as" is replaced with "is".

9). Lines 89-99 have been revised and words/ phrases like "that would" and "under specified…effect" have been deleted.

10). Line 93: Correction of reference with name "Forde" has been made throughout the manuscript.

11). Lines 96-99 have been deleted.

12). Line 104: Location and mode for soil sampling is provided in the revised article in lines 117-118.

12). Line 135: "Indole" is not capitalized and converted to "indole".

13). Line 148: Name of City and State for USA locations are now given in the methodology.

14). Line 169: The name of spectrophotometer is now given in the section for auxin quantification as suggested by the worthy reviewer.

15). Word "sterilizes" replaced with "sterilized".

16). Line 210: Soil mixture ratio revised to 6:1:1.

17). Line 235 and 240 have been moved to methods.

18). Line 246: Start of results for auxin production has been modified and lines 246-248 have been deleted. Similarly, lines 269-271 have also been modified in result section.

19). Line 282: Figure 6 showed the effect of bacterial strains on the root proliferation in the inoculated plants. Line has been modified accordingly.

20). Line 322: "showed stimulatory effects on rosette growth biomass" replaced with "stimulated rosette growth".

21). Line 328: Words changed to "controls".

22). Line 363: Word "parameters" has been deleted.

23). For figure 5 (now figure 7), bar charts have been converted to Box and Whisker plots.

## *Experimental design*

In all tables for pot trials, water treated plants were used as control. Indeed, majority of the bacterial strains recorded promising growth for vegetative parameters over water treated controls. For negative control, bacterial strain was not used as we do not have culture of bacteria that may be deficient for number of plant growth promoting traits.

## *Validity of the findings*

Reviewer's concern with reference to number of replicates is now clarified in the revised article. For 'Microphenotron' bioassay single lane of eight PCR tubes or wells was placed for each strain or control in three replication and experiment was repeated three times. Overall, 72 PCR tubes were used for single strain or B5 media (control). Similarly, for pot trials, experiment was also repeated three times with overall 27 seedlings inoculated for each strains or control. Although, more replicates may be included to increase the authenticity of the experiments. Overall, currently used number of replicates may be sufficient to predict the outcome of the results. Moreover, our results indicated that strains that performed well in 'Microphenotron' bioassay also showed good results in pot trials for wild-type *A. thaliana.*

---

## Round 0.3 · Major Revisions

We appreciate the efforts you have taken to revise the manuscript, but with a strongly split decision we feel it is important that you address the ongoing criticism of the poor grammar and jumbled narrative, as well as address the points raised by the most recent review.

·

Basic reporting

The text requires a number of revisions for format and style. I gave up after Introduction because Abstract and Introduction are jumbled. Abstract should follow the convention of background, unmet need, methods, results, discussion/conclusion. Introduction should follow this outline.
1. Field of study:
a. PGPR play a critical role
b. Variations influence field performance
c. We need to screen lots of them to find ones that work
2. Unmet need – reliable high throughput, low cost PGPR assay
a. PGPR can be characterized on the basis of beneficial biochemical attributes but these in vitro assays do not consistently predict field performance
b. Pot trials are expensive and fraught with bias
c. Large scale robotic phenotyping systems work great but they are expensive
3. Microphenotron system shows promise
a. Works with metabolites
b. Hasn’t been applied to isolates
c. Hasn’t been validated with pot trials
4. In this study we
a. Isolated rhizobacteria
b. Screened them with all of the above
c. Observed that ___

Experimental design

I think it is great that the authors included pot trials in this manuscript; however, the number of replicate pots (three) seems low and I do not understand Figure 7. Methods states pots were inoculated with bacteria. I presume there was some sort of negative control too but those categories are not shown in Figure 7 and even then the authors should show the relationship between the response in the microphenotron versus the response in potted plant trials. I suggest a biplot, with the x-axis a metric derived from microphenotron data (like root growth) and the y-axis a metric (like fresh weight) derived from the potted plant trials. This graph would show if the microphenotron predicted plant growth response.

Validity of the findings

See above

Additional comments

Specific comments on first 65 lines.
Replace bars in Figures 4 and 5 with box/whisker plots like Figure 7. Box is almost always better than bar.
Line 25. Revise to “that uses 96-well”
Line 26. Delete “of Arabidopsis” (I bet you could use other plants).
Line 29. Replace “or” with “and”
Line 30. Format is Genus species. Revise to “Acacia arabica”
Line 30. Replace “The final taxonomic status of bacterial strains was confirmed through” with “The phylogeny of these rhizobacteria was determined by...”

Line 34. Here and throughout, write actively and define strains. Replace “and the highest activity was exhibited by Bacillus endophyticus S-6” with “A strain (S-6) of Bacillus endophyticus exhibited the highest activity”

Line 35. Describe all the methods used in one section (16S sequencing, PGPR traits (ACC deaminase...) UPLC, microphenotron, and pot trials) and then summarize results observed.

Line 38. As above, the reader doesn’t know these strains and write actively. Revise to “Two strains (S-7 and S-11) of Psychrobacter alimentarius procuded the most IAA, ICA and ILA.”

Line 54. Revise to “(PGPR) play a critical role in soil fertility.” Delete “and plant growth promotion”

Line 55. Write actively. Revise to “An incomplete understanding of the mechanisms of plant growth promotion hinders the application....”

Line 57. Delete “the” here and wherever you can.

Line 60. Delete “bacterial metabolites” (ACC demaminase is an enzyme not a metabolite) and replace “has” with “have”

Line 61. Replcae with “does not consistently predict”

Line 62. Greenhouse trials are not an in vitro technique, so you cannot use them as an ex

Line 65. Reorder these sentences to “ It can...in A. thaliana. In the present study...” and move them to the end of Introduction.

Line 69. This paragraph appears to contradict the previous one. You just said in vitro assays do not always produce a desirable outcome.

Line 345. To make it clear who developed the method, revise to “Forde et al. (2013) developed ...

---

## Round 0.4 · Major Revisions

Your manuscript is still challenging from an English language perspective and some of the figures are still confusing. One of the reviewers has commendably offered to provide editorial assistance. This is attached. Please incoporate their edits into your revision.

·

Basic reporting

The manuscript has been improved and remains very interesting to me; however, I have a few significant suggestions. Introduction requires revision, as it doesn’t follow a coherent outline. I suggest the order. To illustrate this structure of revision I have attached a pdf.
1. (Background) PGPR and their applications
2. (Justification) Current screening methods are inadequate
3. Microphenotron

Experimental design

No comment

Validity of the findings

I think I failed to communicate what I mean by “biplot” for Figure 9. I meant that the manuscript needs to show a plot that compares growth response as measure by Microphenotron versus pot trials. That plot would have rosette weight, measured in pot trials, on the x-axis and root length, measured with in Microphenotron, on the y-axis. Each point would correspond to the strains and control and show the variation, in x and y directions, for each variable. That plot would allow readers to determine if the microphenotron and pot trials correlate with each other. To facilitate discussion, I suggest that symbols indicate strain or traits associated with sets of strains.

Additional comments

Line 262. Why cap “Indolic?” Also revise to “….supernatants: indol-..”
Line 280. Do not just tell the reader a figure exists. State the result and direct the reader to the data. Delete “Figure 5 …bioassay.”
Line 282. Again, state the result (All tested strains increased….(Fig. 6).”
Line 288. As above, state result, don’t announce that a figure exists.

---

## Author Rebuttal · Round 0.4

With reference to article #52899 titled "The Microphenotron: A novel method for screening plant growth-promoting rhizobacteria" a revised version of the article is submitted for further consideration and processing. The article has been revised as per reviewer suggestions. Point to point response follow below:

**RESPONSE TO EDITORS' COMMENTS**

We are very thankful to the Editor to provide us with another opportunity to revise this article. All the sections of this article have been revised keeping in view the 'reviewer 3' comments. The introduction, methods, results, and discussion have been modified in the revised version. Figures 4 and 5 have been converted to "Box and Whisker plot" according to the reviewer's suggestion. Moreover, an effort has been made to remove the grammatical errors in this article.

**RESPONSE TO "REVIEWER 3" COMMENTS**

*Basic Reporting*

We are very thankful for the worthy 'reviewer 3' comments to improve the text of this manuscript. Abstract and introduction sections have been revised according to his valuable suggestions. Abstract has been divided into different sections i.e., background, methods, and results/ conclusion.

For the introduction, we have also followed the reviewer's outline for revision. In the first paragraph, we have given the critical role of PGPR in field study and other factors that can influence microbial performance in the field. However, we have retained literature related to the role of phytohormones in plant growth promotion as we have used mutant lines of *Arabidopsis thaliana* impaired in auxin/ ethylene signaling. In the previous revision, we have already discussed the significance of 'Microphenotron' to select effective PGPR. Nevertheless, in this revision, we have also given the importance of 'Microphenotron' in the screening of PGPR for pot trials.

*Experimental design*

For pot trials, the confusion related to the number of replicates is already addressed in the previous revision. For each strain, three pots were placed, and the experiment was repeated three times. In this way, we collected date for 27 seedlings from three experiments. It has been mentioned in lines 278-280 in the revised manuscript. Figure 7 has been revised and converted into 2 figures i.e., 7 and 8. Moreover, information for the respective control treatment is also

given for each growth parameter. Boxplot and whiskers analysis showed the growth response for different vegetative growth parameters.

Moreover, in the discussion section, we have included "Figure 9" to show the comparison growth in 'Microphenotron' and pot trials. We have made a comparison of 'root length' from Microphenotron and 'rosette fresh weight' from pot trials as per the reviewer's suggestion.

*Response to Additional Comments*

1. Bars in figure 4 and 5 have been replaced with Box and Whisker plots according to reviewer's suggestion.

2. Line 25: Phrase is revised to "that uses 96-well"

3. Line 26: "of *Arabidopsis*" is deleted

4. Line 29: "or" is replaced with "and"

5. Line 30: Authority citation is removed, and species name revised to "*Acacia arabica*"

6. Line 30: "The final taxonomic status of bacterial strains was confirmed through" is replaced with "The phylogeny of these rhizobacteria was determined by…".

7. Line 34: Strains have been defined according to reviewer's suggestion. Moreover, "and the highest activity was exhibited by *Bacillus endophyticus* S-6" replaced with "A strain (S-6) of *Bacillus endophyticus* exhibited the highest activity".

8. Line 35: Methods section is added in the revised abstract as suggested by worthy reviewer.

9. Line 38: Sentence has been revised to "Two strains (S-7 and S-11) of *Psychrobacter alimentarius* produced the most IAA, ICA and ILA."

10. Line 54: Phrase is revised to "(PGPR) play a critical role in soil fertility." Moreover, "and plant growth promotion" is deleted from the sentence.

11. Line 55: Sentence is revised to "An incomplete understanding of the mechanisms of plant growth promotion hinders the application…."

12. Line 57: "the" is deleted

13. Line 60: "bacterial metabolites" is deleted and "has" replaed with "have"

14. Line 61: Phrase is replaced with "does not consistently predict"

15. Line 62: Sentences have been amended to remove the confusion about greenhouse and *in vitro* techniques.

16. Line 65: Sentences have been reordered as per reviewer's suggestion.

17. Line 69: Contradiction in these sentences has been corrected.

18. Line 345: Sentence is started with "Forde et al. (2013) developed……"

---

## Round 0.5 · accepted · Accept

Thank you for your extensive efforts in revising and thanks to our reviewers who worked so hard on this manuscript.